# Global Advancement in Pharmacy Services for Mental Health: A Review for Evidence-Based Practices

**DOI:** 10.3390/healthcare11081082

**Published:** 2023-04-11

**Authors:** Mohammed Kanan Alshammari, Nawaf M. Alotaibi, Suroor Nasser Al Suroor, Rami Saleh Al Saed, Aliaa Ali Al-hamoud, Mawahb Ahmed Alluwaif, Mona Awadh Alamry, Norah Mohammed Alshehri, Bashaier Eed Alfaidi, Rand Abdullah Alzahrani, Basil Bandar Almutiri, Yousef Saud Alosaimi, Amal Saeed Alosman, Abdulsalam Awadh Alharbi, Abdulrahman Meshal Alenezi

**Affiliations:** 1Department of Clinical Pharmacy, King Fahad Medical City Hospital, Riyadh 12211, Saudi Arabia; 2Department of Clinical Pharmacy, Faculty of Pharmacy, Northern Border University, Rafha 73213, Saudi Arabia; 3Department of Pharmacy, Maternity and Children Hospital Dammam, Dammam 63400, Saudi Arabia; 4Department of Pharmacy, Khamis Mushait General Hospital, Khamis Mushait 62441, Saudi Arabia; 5Department of Pharmacy, Umluj General Hospital, Ministry of Health, Northern Region, Umluj City 48312, Saudi Arabia; 6Department of Pharmacy, King Khalid University, South Zone, Abha 62541, Saudi Arabia; 7Alrazi Medical Company, Buraydah City 51941, Saudi Arabia; 8Department of Pharmacy, Dr. Sulaiman Al Habib Hospital, Riyadh 14926, Saudi Arabia; 9Department of Pharmacy, King Abdullah Medical City, Makkah 21955, Saudi Arabia; 10Department of Pharmacy, Northern Uorder University, Rafha 91911, Saudi Arabia

**Keywords:** pharmacy services, mental health, global advancement, systematic review

## Abstract

The symptoms of psychiatric infirmities have variability, and selected drug regimens for mental illness are comparatively complex and individualized; therefore, pharmacy services vary with respect to patients, diseases, healthcare settings, community structures, and countries. Clinical pharmacy services for mental health (MH) are continuously being upgraded. A structured search of the literature was performed in the Cochrane, PubMed (Medline), PsycINFO, Google scholar, Scopus, Science Direct, and Springer Links databases. The title and abstract of each retrieved article were evaluated for relevance. To remove uncertainty and ambiguity, the full-text articles were retrieved and examined for relevance. The articles were further assessed on the basis of inclusion and exclusion criteria. Narrative synthesis was performed, creating new categories and relevant subcategories and further subsections. The articles and the results were assessed for quality and bias. Pharmacists have a range of expertise in psychiatric care. The services can be classified as conventional, extended, and advanced pharmacy services. Conventional services include the quality use of medicines in healthcare settings and medication support services in communities that ensure medication adherence. Pharmacists perform extended roles in collaborative medication therapy management, multidisciplinary community mental health teams, collaborative care, patient education, home medication review, hospital-to-home transit, and screening services. In the USA, the role of pharmacists was advanced by prescribing as collaborative and interim prescribers. Australia launched an accredited program for psychiatric first-aid pharmacists. Pharmacists can provide mental care to rural populations using health technology. The role of pharmacists in MH is appreciated either independently or as a team member. Patients and healthcare providers rank the services of pharmacists in MH highly. Still, there is a margin for improvement in the training of pharmacists. Pharmacists cannot provide sufficient time to their patients. Public awareness about the role of pharmacists in MH needs more attention. Moreover, the training of psychiatric pharmacists should be standardized around the world.

## 1. Introduction

More than 1 billion people around the world are suffering from addictive disorders or MH issues. MH causes 7% of the burden of all diseases around the globe. Depression is a major reason for disability, and it is estimated that 10% of deaths around the world are associated with suicide [1]. Major depressive disorder has been the second-largest factor contributing to disability since 2020 [2]. During the last two decades, the incidence of mental disorders in adolescents and children in developed counties has escalated, despite large investments to improve MH services. The insufficiencies of services for mental care are a substantial hindrance in disease management [3]. In the European Union, factors contributing to depression include severe diseases such as cognitive impairment, grip strength, and limitations in daily living. The provided services do not meet the needs of patients [4].

According to the Canadian Mental Health Association (CMHA), more than 0.5 million people miss work each week due to MH issues. This leads to annual economic losses of CAD 51 billion in Canada alone [5]. The Australian Bureau of Statistics estimated that 45% of Australians have suffered from MH issues during their life span [6]. Millions of Americans suffer from mental problems [7]. Among the low–middle-income countries (LMICs), a national survey of MH in India disclosed that 150 million people needed some sort of intervention for good MH [1].

MH issues need proper therapy and care. Therefore, pharmacists are well positioned to develop and improve pharmaceutical care (PC) services. Pharmacists can reduce the risks associated with mental disorders [1,8]. Pharmacists are trained and have the skills to cope with the MH issues of patients. They are assumed to be the most accessible healthcare provider to the public and an initial source of information [8]. Pharmacy services in MH include the review of medicine utilization, providing recommendations for therapy, patient counseling and education, medication therapy management, and providing references for prescribers [9]. These types of interventions have proven impacts on various aspects of treatment, e.g., adherence, reductions in medication duration, reductions in the cost of treatment, and increases in quality of life (QOL) [7,10]. The symptoms of psychiatric problems have variability, and selected drug regimens for mental diseases are comparatively complex and individualized; therefore, pharmacy services vary with respect to patients, diseases, healthcare settings, community structures, and countries. Clinical pharmacy services for MH are continuously being upgraded. A comprehensive analysis of pharmacy services in MH can provide evidence for researchers and policy makers to understand the scope of pharmacists in MH and further improve services. It can also determine the core responsibilities of pharmacists in teams and direct patient care. This review provides a rapid overview of the scope of services and a segmentation of services with reference to transformation, highlights the associated challenges in services, provides an overview of the quality of the provided services with reference to societal and professional perspectives, and concludes with a futuristic approach to services on the basis of results obtained from pilot projects as well as recommendations to strengthen the pharmacy services in MH.

## 2. Methods

### 2.1. Search Strategy

The PRISMA (preferred reporting items for systematic reviews and meta-analyses) guidelines were followed. A structured search of the literature was performed in the Cochrane, PubMed (Medline), PsycINFO, Google scholar, Scopus, Science Direct, and Springer Links databases. The following descriptors were used: psychological interventions by pharmacist, pharmacy services in MH, MH services of pharmacist, MH assistance by pharmacists, community MH services, community/retail pharmacy services in MH, pharmacy practices for MH, PC for MH, pharmaceutical service in MH, clinical pharmacists and MH, clinical pharmacy and MH, pharmacy and psychiatry, neurological issues and pharmacy, mental issues and pharmacy, psychiatric pharmacy, collaborative care in MH, hospital pharmacy services in MH, psychiatric issues solved by pharmacist, and PC for intellectual disability.

### 2.2. Article Selection and Data Collection

The title and abstract of each retrieved article were assessed for relevance. In the case of any ambiguity or uncertainty, the complete article was investigated for relevance. The articles were further assessed according to inclusion and exclusion criteria, as given in Table 1.

The selected articles that explored pharmacy services but did not focus on MH were separated for this study. For the identification of the main services that were depicted in the literature, each article was fully studied. However, a manual search was also performed to retrieve more articles that were cited in different articles but were not found using the search strategy.

### 2.3. Study Selection

The studies identified through the literature search were screened by two independent reviewers to assess their eligibility. Any disagreements were resolved through consensus or by a third reviewer. After evaluating all the databases, the studies were screened for duplications, which were then deleted. The articles were also excluded after screening the titles and abstracts.

### 2.4. Data Extraction

Data were extracted from the eligible studies using a standardized data extraction form. The data included information on study references, designs, research objectives, and outcome measures.

### 2.5. Quality Assessment

The Mixed Methods Appraisal Tool (MMAT) was used to appraise the quality of the empirical studies, as it covers a variety of methodologies. The MMAT includes five core quality criteria for each of the five following categories of study designs: qualitative research, randomized controlled trials, nonrandomized studies, quantitative descriptive studies, and mixed-method studies. Critical appraisals of the methodological quality and the risk of bias assessment of the included papers were undertaken independently by two reviewers (RSA and BEA). A third reviewer (MA) was consulted in cases of disagreement without reaching consensus.

### 2.6. Analysis

A narrative synthesis approach was used to synthesize the findings of the included articles due to the heterogeneity of the studies in the review, with a range of methodologies. First, a preliminary synthesis was conducted to search the studies and present results in a tabular form. Then, the results were discussed by two reviewers (RSA and RAA) and structured into themes. The studies included in the narrative synthesis were then summarized within a framework.

## 3. Results

### 3.1. Literature Retrieval

In total, 48,744 articles were recognized by the search of the databases. Then, 48,604 articles were removed on the basis of ambiguity and duplication. On the basis of the inclusion and exclusion criteria, 86 articles were removed. Thus, 68 articles were included in this review (Figure 1).

### 3.2. Characteristics of Articles

The 68 articles in this review include 6 retrospective studies, 22 RCTs, 1 simulated patient, 5 qualitative studies, 6 mixed-method studies, 12 descriptive studies, 2 observational studies, 4 prospective studies, 5 cross-sectional studies, 2 controlled studies, and 1 uncontrolled feasibility study.

### 3.3. Risk of Bias and Quality Assessment

All the included articles met the screening criteria: having clear research questions and addressing the research questions based on the collected data. Almost all articles were rated as average or above-average quality (see Appendix A).

### 3.4. Pharmacy Services

According to the nature of pharmacists’ roles in MH, pharmacy services were divided into three types, i.e., conventional, extended, and advanced pharmacy services.

#### 3.4.1. Conventional Services

Pharmacists traditionally perform duties in different segments of MH, i.e., the quality use of medicines in healthcare settings, medication support services in communities, and medication adherence (Table 2).

#### 3.4.2. Extended Services

Commonly performed extended services include residential medication management reviews(RMMRs) and home medicine reviews(HRMs) in Australia, medicine use review services (MURs)in the UK, medication therapy management (MTM) in the US, and medication review services (MRS) in New Zealand [20] (Table 3).

#### 3.4.3. Advanced Services

Pharmacists are becoming more and more engaged in MH, and their role is advancing. The nature of the advancement varies from country to country (Table 4).

### 3.5. Outcomes of Pharmacy Services and Prospects

#### 3.5.1. Healthcare Provider Satisfaction

Two studies from Australia elaborated the role of psychiatric-specialist pharmacist (PSPs). There was a great recognition and acceptance by other HCPs for their role in the management of psychiatric patients. PSPs addressed an unmet need for pharmaceutical services. The medication expertise of PSPs was highly regarded [40,62]. A recent study from Qatar revealed high expectations of doctors and nurses for the tasks of pharmacists at a mental care hospital. They had positive perceptions about the clinical role of pharmacists. On the other hand, conventional clinical services performed by pharmacists were more favorably analyzed than advanced clinical responsibilities such as prescribing and medication management [63].

#### 3.5.2. Patient Satisfaction

Black and his colleagues reported that victims of mental issues noticed the significance of pharmacy services. Conventional pharmacy services such as medicine information were considered more essential than clinical services. This study also found that the significance of a pharmacy service was always associated with its provision [64].

Two studies from Australia viewed the customers of pharmacies in community settings. The confidence levels of patients were assessed. The services were highly valued by participants, and they considered the community pharmacies to be safe places to obtain advice on MH and wellbeing. The studies showed good conjecture regarding the role of pharmacists [65,66]. However, the rural population of Australia wished to find expanded services for MH similar to the urban population [67]. In the US, schizophrenic patients viewed pharmacists as knowledgeable sources, but their perception of pharmacists was primarily as dispensers of medicines [68].

### 3.6. Challenges and Limitations

Only two challenges were found in the literature, i.e., related to practice and pharmacist training.

#### 3.6.1. Challenges of Practice

A lack of time with pharmacists, low awareness in the public about the role of pharmacists, and the low knowledge of customers about provided help and available resources are major challenges, especially for the depression screening services at community pharmacies [69]. The literature indicates challenges in the implementation of an accredited first-aid program [70]. Challenges regarding training in first aid were also reported [71]. In addition, some challenges such as a lack support from the owner, manager, or staff of a pharmacy; privacy limitations; and time constraints were reported from community pharmacies for the provision of medication management in MH [39]. Key challenges identified in patient care in the hospital-to-home transition included limited knowledge and insufficient communication. It was also reported that there is a need for a standardized role [53].

#### 3.6.2. Inappropriate Training of Pharmacists

Suboptimal attitudes towards disease conditions and a lack of self-reliance in the provision of clinical services to patients necessitate special didactic approaches. It was also identified that educational programs should shift from the conventional focus of therapeutics such as antipsychotic remedies. The adaptation of evidence-based medication and practices will decrease the stigma of MH. It was also noticed that an improvement in the professional confidence of pharmacists is needed to offer suitable MH services. The training of pharmacists must have a focus on the development of communication skills [72,73]. A recent survey in the US reported the views of pharmacists about their training, and they claimed that the emphasis on MH in their training was not adequate [74].

## 4. Discussion

Medicines remain a key modality for the treatment of numerous psychological issues such as bipolar disorder, depression, and schizophrenia. Therefore, it is logical that pharmacists are supposed to contribute to the management of mental diseases via the quality use of medicines. The literature highlights this imperative clinical task of pharmacists in hospital and community settings [75,76].

### 4.1. Quality Use of Medicines in Healthcare Settings

A study from Canada and three other studies from the USA also showed the potential of community pharmacies to encounter the misuse of illicit drugs for psychiatric issues. It was also found that the proper care of patients can eradicate addiction [13,14,15,16]. However, a study of an Australian community pharmacy showed the medication-centered approach for antidepressant use. However, this study lacked patient-centered communications. The behavioral and psychosocial discussions, particularly those related to lifestyle modification, were missed. Emotional empathy building and facilitating the involvement of patients were also overlooked [17].

Some previous analyses stated that patient counseling by pharmacists and the monitoring of therapy can optimize medication and improve adherence in hospital and community pharmacy settings [75,76].

### 4.2. Medication Support Service in Community

An exploration in a primary care setting in Australia revealed the decision making via a deep discussion between community pharmacists and physicians to cure mental disorders. A detailed conversation about issues associated with potential and actual medication was carried out. The approaches that optimize individual medication regimens were the focus. Face-to-face conference meetings were conducted for 44 cases.

The doctors assumed their final decision-making task, but it was predicted by them that pharmacists are capable of medication adherence improvement. The physicians also acknowledged that a number of patients were more interested in information sharing with pharmacists than doctors [18].

Similarly, another qualitative study from Australia showed that pharmacy staff are ideally positioned to implement a medication support service [19].

### 4.3. Medication Adherence

Nonadherence to psychotropic therapy remains a common issue. Among them, nonadherence to antidepressant therapy was high. Treatment adherence prevents the recurrence of depression and decreases the healthcare costs [77]. Many studies have evaluated the success of complex and multifaceted interventions that improve adherence to antidepressants. These interventions included the strategies of patient education, telephonic follow-up to assess the patients’ progress, and the feedback of a health team [78].

The engagement of pharmacists in medicine dispensing is historically understood. They are ideally positioned to support patient therapy and collaboration with patients. A pharmacist can easily evaluate and promote the importance of adherence to therapy. Previously, many studies were conducted in Australia, Brazil, Spain, the USA, the Netherlands, Saudi Arabia, and Kuwait with pharmacists involved in the management of depressive patients; however, the conclusions from these studies varied but showed measurable differences [22,23,24,25,26,27,28,29,30]. In contrast to these findings, the analysis by Brown et al. did not show proof of a difference between the usual and pharmacy-based management of depression in adults for symptomatic relief [78].

However, the analyses in some studies showed significant [22,32] or marginally significant [20,24,28] improvements in therapeutic adherence. Here, it is notable that interventions by pharmacists significantly improved the adherence to antidepressant therapy (OR = 1.64, 95% CI = 1.24–2.17) [30].

One study from the USA evaluated a pharmacy-based intervention for patients with serious mental disorders such as bipolar disorder, schizophrenia, and schizoaffective disorder. It showed an improvement in adherence to antipsychotic medication [33]. Although the facts point out the significance of pharmacists in improving psychotropic medicine adherence, these practices are not frequently used. A simulated client study from Australia indicated that the counseling of community pharmacists for antidepressants is not outstanding [34].

### 4.4. Collaborative Medication Therapy Management

A model of joint care known as collaborative drug therapy management (CDTM) is actually an accord between a pharmacist and physicians. In this type of effort, a pharmacist assumes responsibility for complete patient assessments, the selection and adjustment of drug regimens, and the monitoring and follow-up of patients for drug therapy. A descriptive study demonstrated the protocols for out-patient clinical pharmacy services by engaging a board-certified psychiatric in-community treatment center for substance abuse disorders and MH problems. This study also found certain obstacles in the USA, including prescriber–pharmacist relations, a lack of state and federal laws, the development of pharmacy informatics, and pharmacy services billing [35]. The high recognition of pharmacist advice in the US reveals the integration of pharmacists in CDTM. Integrating CDTM with a patient-centered medical home is an estimable patient-centered approach that addresses the therapeutic issues of homeless people [36]. Similarly, another study showed the development of a pharmacist–psychiatrist CDTM clinic [37]. Moreover, the pharmacy residency of a homeless-patient-aligned care team improved psychiatric pharmacotherapy follow-up, and interventions were found to be very effective [38]. On the other hand, a mixed-methods study of community pharmacies in Australia revealed some opportunities and challenges in the execution of an intervention that related to the therapeutic management of MH [39].

### 4.5. Multidisciplinary Community Mental Health Teams

After a long debate over many years, only a few studies assessed the integrated work of pharmacists in teams of professionals. The study from Australia by Bell et al. showed the incorporation of pharmacists as members of a community MH team, where five pharmacists were included for one day per week (for a 6-month duration). Their offerings were welcomed and valued by the team. The perspective of MH professionals and pharmacists is that the inclusion of pharmacists in community MH teams (CMHTs) addresses a dire need for pharmacy services among customers and staff. However, participation for only a single day per week seems trivial and hinders the impactful collaboration between the pharmacists and the CMHT members. However, this study answered the key question of whether a pharmacist should be considered an essential member of an interdisciplinary MH team [40].

Another study in the US by Mathys et al. investigated the inclusion of pharmacy students in multidisciplinary MH teams. The results showed no considerable differences between the control and experimental groups. However, the measures indicated comparatively high rates of patient counseling, interventions, and medication reconciliation [41].

A randomized control trial (RCT) in Flanders, Belgium was conducted by Liekens and colleagues. This RCT evaluated the impact of a pharmacist’s training regarding the counseling of patients that initiate antidepressant therapy. It was concluded that special training on controlling depression improved the quality of the interactions between pharmacists and patients. This study also portrayed an improvement in lifestyle. It also revealed that pharmacists addressed psychosocial issues using PC [42].

A study from the US also described an interprofessional collaboration at an attention deficit hyperactivity disorder (ADHD) clinic for adult care. The execution of a collaborative service at the ADHD clinic demonstrated a successful coalition for MH [43].

Wang et al. conducted a study in the US that verified the role of pharmacists in CMHTs. They assessed the impacts of services provided by a psychiatric pharmacist in the health center. The community had limited access to psychiatrists. The study showed that the psychiatric pharmacist contributed to various areas of care such as psychotherapy and specialty care [44].

### 4.6. Collaborative Care and Patient Education

The partnership of healthcare providers and pharmacists has shown an affirmative role in juridical settings. A study from a prison in France showed that monthly meetings among pharmacists and psychiatrists about the execution of guidelines regarding benzodiazepine use led to significant decreases in the benzodiazepine doses per day [45].

A study from Sweden by Schmidt et al. showed that monthly multidisciplinary meetings by pharmacists decreased the use of antipsychotics, non-recommended antidepressants, and non-recommended hypnotics [46]. Bower and colleagues found an improvement in medication adherence associated with antidepressant use due to collaborative care strategies in primary care settings (OR = 1.92, 95% CI = 1.54–2.39) [64]. Two other studies from primary care settings in the US also found effective collaborative roles of psychiatric pharmacists [48,49].

A recent study from the Department of Veterans Affairs (VA) of the US showed the successful integration in practice and suggested that foundational concepts for the integration of pharmacists into interprofessional care teams can be applied in non-VA settings. MH clinical pharmacy specialists can work as direct patient care providers. Their services also improve clinical outcomes, access to care, and safety [50].

A study from India in a tertiary care setting for schizophrenia also showed that collaborative patient education by pharmacists and psychiatrists improved medication adherence and QOL [51].

### 4.7. Home Medication Review and Hospital-to-Home Transit

A study from the US concluded that pharmacists’ contributions in care coordination promote adherence. They also manage serious mental illness [53]. Home medication review and hospital-to-home transit are two important aspects of PC in the community. Nishtala et al. found the significance of educational interventions and medicine use reviews for psychotropics in residential geriatric care communities. The hypnotic use was considerably decreased by medicine use reviews, as proven by a pooled odds ratio of 0.57 (95% CI: 0.4–0.7) [52].

Two Australian studies used the drug burden index (DBI) to determine the outcomes of RMMRs and HRMs that were conducted by pharmacists. These studies aimed to examine the total sedative medicine and anticholinergic burdens of people. Hence, these studies demonstrated significant reductions in the exposure to sedative and anticholinergic medicines of residents [54,55].

Nishtala and colleagues conducted a retrospective study of the DBI. A study of 500 HRMs in 62 geriatric care centers confirmed that HRMs conducted by pharmacists led to reductions in median post-RMMR scores (from 0.5 to 0.3) [54]. Similarly, another retrospective study was conducted in an Australian community. In this study, 372 HMRs were conducted by 155 pharmacists, and the mean DBI score was significantly reduced (from 0.5 to 0.2) [55]. A recent cross-sectional national Irish study of a pharmacy claims database also showed similar results in elderly people [56].

In Australia, Gisev and colleagues used a panel of experts to evaluate medication reviews that were independently conducted by pharmacists. This study showed that the recommendations and findings of pharmacists were appropriate [65].

### 4.8. Screening Services

Pharmacists are easily accessible in community settings, and no prior appointment is needed to screen for depression. They are considered trustworthy by the general public [20]. Earlier studies demonstrated that well-trained and competent pharmacists are equipped with the knowledge and skills to help the general public identify the early symptoms of depression, and they can support their customers during treatment [18,66,67,68,72].

However, the screening of depression in a community pharmacy setting is not a common practice. Some pilot studies for feasibility showed that community pharmacists have the capability of early diagnosis. They are well positioned as a referral service for people at risk of depression [69,79,80,81].

A study from the US also showed that pharmacists are ideal resources for monitoring patients that have a risk of suicide [82]. A recent qualitative study from the US also showed that patients and prescribers think that community pharmacists have the ability to screen patients for depression [83].

### 4.9. Collaborative Prescribing

An Australian study of secondary care investigated the attitudes of HCPs and customers about pharmacists’ role as “collaborative prescribers” in MH. Both HCPs and customers recognized the role of pharmacists and PC in MH [84].

Joint efforts between doctors and pharmacists in the field of MH were evaluated in an Australian study of a primary care center. A total of 44 cases were scrutinized, and strategies were briefly discussed. In these cases, the optimization of therapy and treatment strategies as well as the exchange of patient and medical information were carried out. No biases or conflicts were reported [19].

### 4.10. Accredited First-Aid Provider

In 2001, Australia recognized the role of community pharmacists in MH as first-aid providers and set some standards for accreditations. Some basic training was necessary to meet the eligibility [20,71]. After that, it became an admirable training program in many other developed countries and the US. MH first aid is provided to a patient who is suffering from an MH problem or an MH crisis until proper professional help is provided or the crisis is resolved [73,85].

In 2018, Chawdhary et al. found that some challenges are associated with its proper implementation [70]. However, a recent study from the US indicated that patients are comfortable discussing mental issues with trained MH first aid providers. This revealed an opportunity for pharmacists to advance their services for MH [74,86].

### 4.11. Interim Prescribers

Turnover in psychiatric emergency services leads to fluctuations in the availability of prescribers. Patients face many problems due to the discontinuity of care. A retrospective cohort study was performed in the US with outpatients who lost their MH prescriber due to a transfer or turnover and needed medication therapy management (MTM). This study revealed a role of pharmacists as interim prescribers. Moreover, a reduction in the patient flow in psychiatric emergency was found [75].

### 4.12. Clinical Telehealth

Patients from rural areas have difficulties traveling and cannot seek proper MTM. Therefore, a recent study in the US evaluated the functions of a pharmacy MH clinical video telehealth (MHCVT) clinic. The patients showed great satisfaction with these MHCVT clinics. The visits to the clinics resulted in cost savings [76].

### 4.13. Solo PC and Health-Related Quality of Life

PC is assumed to be a direct intervention between the patients, HCPs, and pharmacists. PC optimizes therapy and reduces pessimistic outcomes related to medications. It also contributes to an improvement in health-related quality of life (HRQOL). A study from Colombia explored the execution of a PC program on the HRQOL of epileptic women. The results showed that the application of the PC program notably improved HRQOL [77].

Another study on long-term hospitalized patients with schizophrenia from Serbia also found that PC is a tool that improves the practices of medicine prescription in LMICs, regardless of a physician–pharmacist collaboration [78].

### 4.14. Specialist Mental Health Pharmacy Teams

The medication management of serious patients with mental chaos, such as schizophrenia or bipolar disorder, is not easy. A study from the UK showed the role of a specialist MH clinical pharmacy team that contributed to dose adjustment or polypharmacy and the monitoring of physical conditions and drug adherence. The staff of the surgical ward referred issues to the pharmacists for review.

The specialist MH clinical pharmacy team resolved the queries of the surgical staff about psychotropic medication. This team quickly resolved issues of drug errors and reconciled medication [87].

## 5. Conclusions

Pharmacists are providing many services in MH. Early screening and interim prescribing services will be beneficial and need universality. The training of psychiatric pharmacists in LMICs and developed countries should be standardized.

## Figures and Tables

**Figure 1 healthcare-11-01082-f001:**
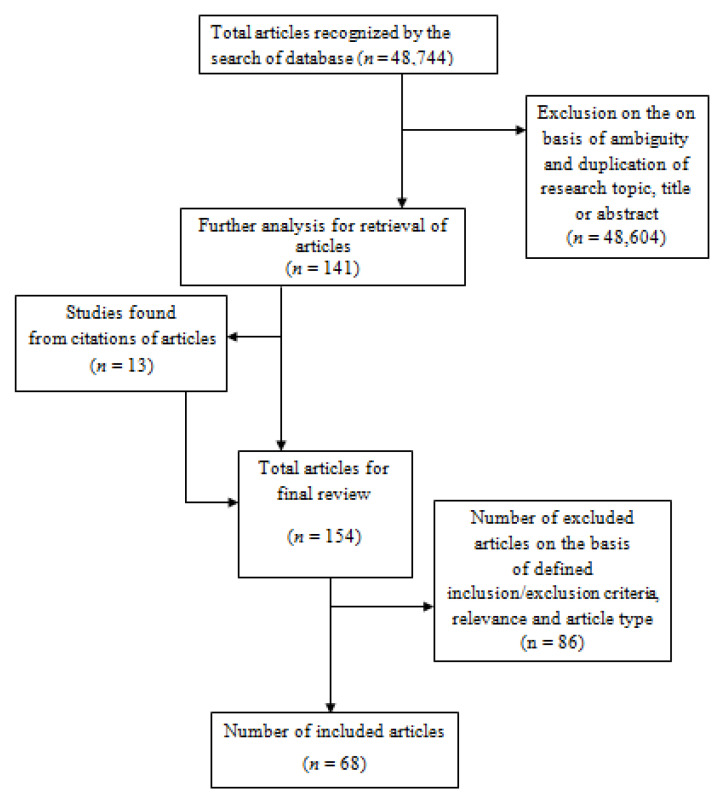
Study selection flow diagram.

**Table 1 healthcare-11-01082-t001:** Study exclusion and inclusion criteria.

Exclusion Criteria		
No.	Category	Criteria
1	Language selection	Articles not written in English
2	Nature of publications	News reports, editorials, blogs, commentaries, opinions, correspondence, research protocols, articles in non-peer-reviewed journals, articles in nonindexed journals, and minireviews.
**Inclusion Criteria**		
1	Duration	Publications from 1990 to present.
2	Publication classification	Full-text research articles in peer-reviewed scientific journals.
3	Language selection	All articles published in English.
4	Healthcare setting	Community, geriatric, hospital, and residential pharmacy services.
5	Methodology	Original articles in peer-reviewed journal investigating/comparing pharmacy services in MH.

**Table 2 healthcare-11-01082-t002:** Summary of studies’ characteristics (conventional services).

Reference	Origin	Study Design	Study Setting	Sample Size	Major Outcomes/Activities	Conclusions/Suggestions
**Quality use of medicines in healthcare settings**						
Lauren, 2020 [11]	Canada	Retrospective analysis	Community pharmacy	∘ 2055 activities; ∘ 1144 patient-care sessions.	Pharmacists were engaged in: ∘ Medication management; ∘ Patient education; ∘ Collaboration in medicine use; ∘ Social support; ∘ Self-care.	Medicine management per session was poor.
Cochran, 2003 [12]	US	Randomized clinical trial	Community pharmacy	46 patients	Feasibility to reduce opioid misuse by the community pharmacy services.	Integrated behavioral intervention decreased opioid misuse.
Natasa, 2006 [13]	Australia	Descriptive study	Community mental health service	56 patients	The pharmacists reviewed: ∘ High-dose antipsychotics; ∘ Antipsychotic poly-pharmacy; ∘ Atypical polypharmacy; ∘ Subtherapeutic doses of mood stabilizers; ∘ Prescribing patterns of hypnotics; ∘ Therapeutic duplication for psychiatric medicines.	The routine reviews of every prescribed drug for patients of community MH may have minimized the risk of ADRs or interactions related to therapy.
Cochranab, 2017 [14]	US	Cross-sectional survey	Community pharmacy	333 patients	Individuals within the mental health subgroup had an increased risk for opioid medication misuse.	Scheduled screening for opioid prescription filling necessitates evidence-based interventions.
Bell, 2006 [15]	Australia	Descriptive study	Community pharmacy	49 people	Pharmacists made 360 recommendations, and 90% of the recommendations were accepted by GPs. Pharmacist were involved in: ∘ drug information provision; ∘ assessment of drug adherence.	Identification of a high prevalence of medicine-related issues.
**Medication support services in communities**						
Hassell, 1997 [16]	UK	Qualitative study	Community (home)	44 telephonic interviews	Pharmacists have roles in: ∘ The treatment of minor ailments; ∘ Referring to GPs or the preference of GPs.	Advisory role of the pharmacist.
Bell, 2007 [17]	Australia	Case conferences	Community (home)	44 people	∘ Medical information was exchanged between physicians and pharmacists. ∘ Therapy recommendations and findings were presented by pharmacists.	Responsibility to share information and discuss treatment options with the primary care physicians.
**Medication adherence**						
Valera, 2013 [18]	Spain	Randomized clinical trial	Community pharmacy	179 patients	Patients that received the intervention showed more adherence during the follow-up of 3–6 months.	Pharmacists’ intervention helped to improve the overall measure of patient wellbeing.
Adler, 2004 [19]	US	RCT	Primary care	533 patients	∘ The 6-month antidepressant use rates for intervention patients exceeded controls (57.5% vs. 46.2%, *p* = 0.03). ∘ The intervention was effective in improving antidepressant use rates for patients not on ADs at enrollment (32.3% vs. 10.9%, *p* = 0.001).	Pharmacists significantly improved antidepressant use.
Brook, 2002 [20]	The Netherlands	RCT	Community pharmacy	148 patients	∘ Patients that received the intervention showed a better attitude than the control group (*p* = 0.03). ∘ Pharmacist coaching was evaluated positively.	Instruction of community pharmacists is an effective tool to improve medication behavior.
Capoccia, 2004 [21]	US	RCT	Primary care	74 patients	Pharmacist interventions at 3 months improved the symptoms of depression and QOL, and these improvements were maintained for one year.	Pharmacist interventions increased antidepressant adherence.
Crockett, 2006 [22]	Australia	RCT	Community pharmacy	106 patients from 32 community pharmacies	The study indicated thatadherence to medicationsimproved wellbeing.	The involvement of pharmacists is beneficial.
Aljumah, 2015 [23]	Saudi Arabia	RCT	Hospital pharmacy	239 patients	The pharmacist intervention directly to patients showed: ∘ More medication adherence; ∘ Therapy-related satisfaction.	The pharmacist intervention improved adherence to medications and other patient-reported outcomes.
Al-Saffar, 2008 [24]	Kuwait	RCT	Hospital pharmacy	150 patients	Patients said that written information (leaflets) and drug counseling (verbal)by a pharmacist had good quality and were easy to understand.	Pharmacists in mental hospitals can play a significant role in the satisfaction of patients.
Finley, 2003 [25]	US	RCT	Hospital pharmacy	125 patients	Pharmacist services had: ∘ Higher drug adherence rate; ∘ Patient satisfaction; ∘ Insignificant clinical improvement.	Clinical pharmacists had impacts on numerous aspects of patientcare.
Valenstein, 2011 [26]	US	RCT	Hospital pharmacy	118 patients	Intervention had impact on antipsychotic medication possession ratios (*p* < 0.0001).	The pharmacy-based intervention increased adherence to antipsychotic therapy in seriously ill patients with mental disorders.
Chong, 2013 [27]	Australia	Simulated patient method	Community pharmacy	20 community pharmacies	Pharmacies provided: ∘ Educational messages; ∘ Adherence-related messages; ∘ Medication adherence monitoring. ∘ The patients were referred back to their prescribing doctors.	Information about the risks and benefits of antidepressants was provided by pharmacists.

**Table 3 healthcare-11-01082-t003:** Summary of studies’ characteristics (extended services).

Reference	Origin	Study Design	Study Setting	Sample Size	Major Outcomes/Activities	Conclusions/Suggestions
**Collaborative medication therapy management**						
Moczygemba, 2011 [28]	US	Descriptive study	Special mental health clinics	695 patients	In total, 194 out of 217 (89%) recommendations by pharmacists were accepted.	Collaborative therapy management integrating pharmacy services isa valuable strategy that addresses drug-related issues.
Tewksbury, 2017 [29]	US	Qualitative study	Medical centers	15 healthcare providers (HCPs)	Collaborative medication therapy management ensures: ∘ The monitoring of atypical antipsychotics; ∘ Medication reconciliation by patient profiling; ∘ Medication screening. Pharmacists’ qualifications justify and have impacts on outcomes.	In psychiatric medical centers, pharmacists can perform collaborative practices.
Tallian, 2012 [30]	US	Mixed-method study	Outpatient psychiatric services	68 patients	∘ Average of 26 min/visit; ∘ USD 4.82 cost/min	In psychiatry health clinics, pharmacists have the ability to collaborate with psychiatrists for patient care.
Pauly, 2018 [31]	US	Mixed-method study	Special mental health clinics	40 patients	Interventions of residential pharmacists include: ∘ The identification of administrative errors; ∘ Adjustments to therapies; ∘ Education and counseling; ∘ Poly-pharmacy reduction; ∘ Referrals for additional services. ∘ Total cost savings = USD 33 613.67	Interventions improve the outcomes of psychotherapy, and there are cost savings.
Hattingh, 2017 [32]	Australia	Mixed-method study	Community pharmacy	163 pharmacists and pharmacy staff	∘ Opioid substitution services; ∘ Relationships with mental health services; ∘ The existence of challenges and opportunities.	Collaborative care improves psychiatric pharmacotherapy.
**Multidisciplinary community mental health teams**						
Bell, 2007 [33]	Australia	Descriptive study	Community mental health teams	Five pharmacists	∘ An appreciation of information provided for non-psychotropic treatment; ∘ Participation in meetings of clinical teams for medication review; ∘ Recommendations and findings.	Pharmacists’ inclusion as members of community health teams ensures the provision of valuable pharmacy services that are needed.
Mathys, 2015 [34]	US	Retrospective study	Community mental health teams	526 patients	∘ Medication reconciliations in the medical chart; ∘ Patient counseling.	Additional interventions by interdisciplinary teams.
Liekens, 2014 [35]	Belgium	RCT	Community pharmacy	40 community pharmacies	∘ Additional educational interventions; ∘ Counseling statements; ∘ Patient lifestyles; ∘ Psychosocial concerns; ∘ Medical conditions; ∘ Therapeutic regimens; ∘ Socio emotional concerns.	The quality of care for depressive patients is increased by pharmacist training.
Casey, 2020 [36]	US	Mixed-method study	Primary care	914 patients	∘ Telephone follow-ups every 2 weeks; ∘ Cost savings of USD 761,280.	A successful coalition of mental health, pharmacy, and primary care.
Wang, 2011 [37]	US	Observational study	Community health center	74 patients	∘ Chart review; ∘ Regular follow-up; ∘ Referrals; ∘ Drug therapy problems.	Valuable contribution of the psychiatric pharmacist.
**Collaborative care and patient education**						
Lerat, 2010 [38]	France	Retrospective study	Prison	473 patients	The benzodiazepine daily dose in the intervention group was reduced to 34 mg compared to the control group (46 mg). Similarly, with mental disorders, the ratio was 30 mg vs. 48 mg. On the other hand, the results for nonopioid therapy were 31 mg vs. 44 mg, while the results for buprenorphine therapy were 63 mg vs. 58 mg. For a non-antidepressant regimen, the results were 29 mg vs. 41 mg, and for an antidepressant regimen, the results were 38 mg vs. 53 mg.	The daily dose of benzodiazepines was reduced after a monthly guideline meeting of psychiatrists and pharmacists.
Schmidt, 1998 [39]	Sweden	RCT	Nursing homes	1854 residents in 33 nursing homes	Significant reductions in the prescribing of ∘ Benzodiazepine hypnotics (−37%); ∘ Psychotics (−19%); ∘ Antidepressants (−59%).	The integration of pharmacists improves the prescription of psychotropic drugs.
Silvia, 2020 [40]	US	RCT	Primary care	141 patients	∘ A reduction in treatment duration (*p* < 0.001) to 31.3 days in the study group compared to 104.5 days in the control group. ∘ Highly satisfied patients (mean points 26.8/28).	The introduction of psychiatry pharmacists in primary care has a constructive effect and improves the treatment of depression.
Chung, 2011 [41]	US	Descriptive study	Primary care	34 patients	Types of interventions include: ∘ Initiating drug therapy; ∘ Adjusting dosages; ∘ Discontinuing ∘ Drug therapy; ∘ Providing medication education.	A psychiatric pharmacist collaboration can improve access to MH services.
Moore, 2020 [42]	US	Descriptive study	Outpatient	900 MH providers	An MH clinical pharmacist improves: ∘ Access to care; ∘ Clinical outcomes; ∘ Patient safety.	A clinical pharmacy specialist improves overall care.
Mishraa, 2017 [43]	India	RCT	Tertiary care	23 patients	∘ Medication adherence improved from 0.7 ± 0.67 (control group) to 1.75 ± 0.2 (intervention group). ∘ Similarly, HRQOL was improved from 16.12 ± 1.98 (control group) to 24.17 ± 0.3 (intervention groups).	Collaborative care by a psychiatrist and a pharmacist notably improved patients’ adherence to medication and HRQOL.
**Home medication review and transition from hospital to home**						
Abrahama, 2017 [44]	US	Qualitative study	Transition from hospital to community(homes)	6 patients + 16 HCPs	Transition from hospital to community needs pharmacist consultation for: ∘ Hospital administration; ∘ Discharge scheduling; ∘ Outpatient care.	Serious patients with mental diseases adhered to therapy by increasing the pharmacists’ participation in the care coordination process. It also promoted the optimal management of diseases.
Nishtala, 2009 [45]	Australia	Retrospective study	Aged-care homes	500 residents in 62 aged-care homes	∘ The recommendations of pharmacists to GPs significantly decreased the mean DBI score (*p* < 0.001).	In geriatric care, medication reviews by pharmacists reduced the rate of anticholinergic and sedative prescription.
Castelino, 2010 [46]	Australia	Retrospective study	Community	372 HMRs by 155 pharmacists	A reduction in mean DBI score (from 0.5 to 0.2)	HMRs by pharmacists in geriatric care reduced the burden of anticholinergic and sedative prescriptions.
Gisev, 2010 [47]	Australia	Descriptive study	Community	209 medication reviews	∘ The panelists agreed to76% of the findings. ∘ In total, 81% of the recommendations were correct. ∘ In total, 69% of the recommendations were applicable. ∘ In total, 77% of the reviews had the potential fora positive clinical outcome.	According to HCPs, pharmacists’ recommendations and findings optimize therapy.
**Screening services**						
Hare, 2008 [48]	US	Feasibility (uncontrolled study)	Community pharmacy	18 participants	Fourteen out of eighteen participants had symptoms of MDD.	With training, community pharmacists are capable of performing screenings.
Knox, 2006 [49]	US	Uncontrolled study	Pharmacy on university campus	25 participants	In total, 64% of the participants rated the pharmacist’s screening tool as very valuable.	Depression screening by pharmacists is viable.
Rosser, 2013 [50]	US	Prospective study	Community pharmacy	3726 patients	Treatment of psychiatric patients was modified or initiated.	Pharmacists have the ability to identify undiagnosed patients with the symptoms of depression.
O’Reilly, 2014 [51]	Australia	Feasibility (uncontrolled study)	Community pharmacy	42 screenings	About 70% of patients were referred to a psychologist or a GP for further assessment.	Pharmacists have the ability to screen and assess the risk of depression.
Gillette, 2020 [52]	US	Cross-sectional survey	Community pharmacy	225 participants	Patients with suicidal thoughts were identified by community pharmacists.	Pharmacists need further training and educational programs for suicide prevention.
Mospan, 2020 [53]	US	Qualitative study	Community pharmacy	12 patients and 4 HCPs	Three key themes: (1) privacy and confidentiality in screening; (2) the potential of pharmacists’ interventions; and (3) MH stigma concerns.	HCPs and patients believe that community pharmacists are able to screen depressive patients.

**Table 4 healthcare-11-01082-t004:** Summary of studies’ characteristics (advanced services).

Reference	Origin	Study Design	Study Setting	Sample Size	Major Outcomes/Activities	Conclusions/Suggestions
**Collaborative prescribing**						
Wheeler, 2012 [54]	Australia	Qualitative study	Secondary care	HCPs (n = 9) and patients (n = 3)	Discussed solutions included: ∘ Building relationships; ∘ Robust training and competency assessment; ∘ Framework of collaborative prescription model.	Receipt and gratitude regarding collaborative prescription model with the role of specialist pharmacist.
**Accredited first-aid provider**						
Hadlaczky, 2014 [55]	Sweden	Meta-analysis	Multiple MH care units	15 studies	MHFA decreases negative feelings and increases awareness about MH and caring behaviors.	Pharmacist engagement in MHFA
Dollar, 2020 [56]	US	Cross-sectional survey	Community pharmacy	358 participants	High level of comfort for patients (*p* < 0.01). Public satisfaction with the qualifications of pharmacists (*p* < 0.01)	MHFA training could increase public satisfaction and comfort.
**Interim prescribers**						
Gibu, 2017 [57]	US	Retrospective cohort study	Hospital	81 patients	∘ A number of interventions. ∘ Outcome assessment: in total, 80% of prescriptions were renewed.	Emergency visits by patients were decreased by pharmacist interventions.
**Clinical telehealth**						
Leach, 2016 [58]	US	Prospective study	Psychiatric hospital	19 patients	∘ Saved 34 miles of travel/visit (for patients); ∘ Saved USD 674.50; ∘ Satisfaction: 100%.	MHCVT by a pharmacist is useful, helpful, and easily applicable.
**Solo pharmaceutical care and health-related quality of life**						
Ilickovic, 2016 [59]	Serbia	Prospective study	Psychiatric hospital	49 patients	The 182 interventions suggested by a clinical pharmacist (70% at the drug level) led to the discontinuation of therapy; the modification of dosage and dose; and the addition of a new medicine. The majority of interventions were accepted by physicians.	PC improves prescribing practices and modifies therapy.
Losada-Camacho, 2016 [60]	Colombia	RCT	NGO-organized outpatients	182 patients	∘ The HRQOL of epileptic patients increased from 2.61 (*p* < 0.001) to 12.45 (*p* = 0.072).	PC notably improves HRQOL.
**Specialist mental health pharmacy teams**						
Raynsford, 2020 [61]	UK	Retrospective study	Primary and secondary care	316 patients	∘ In total, 23 (7%) records were upgraded (missing clozapine). ∘ Clarification of information about patient discharge. ∘ High-dose revision. ∘ Multiple prescriptions of antipsychotics. ∘ Error correction. ∘ Adherence issues were probed. ∘ Following up for missed health checkups.	Specialist teams could quickly resolve medication problems. Leads to bridging between secondary and primary care.

## Data Availability

Data is available in the Appendix A.

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
