# Peer review of "Global Advancement in Pharmacy Services for Mental Health: A Review for Evidence-Based Practices"

_healthcare, 2023, doi:10.3390/healthcare11081082_

Round 1

Reviewer 1 Report

Thank you for the opportunity to review this paper- it is an interesting paper on an important topic. However, significant improvements are needed for it to be at a standard suitable for publication. These include: 

* Improved use of English language. The paper contains a number of spelling and grammatical errors and was very difficult to read in parts 

* More synthesis and analysis in both the introduction and results. At the moment, both sections largely consist of lists of relevant aspects of other papers, without an overall arching story. 

* More detailed description of methods. For example, what specific search terms and combinations of these were used? Which authors screened and analysed which papers? How exactly were papers analysed? 

* Conclusions and recommendations do not flow logically from the results. Please make the ideas in your conclusion more explicit in the results section of the paper. 

I look forward to reading the next version of this paper! 

Author Response

* Improved use of English language. The paper contains a number of spelling and grammatical errors and was very difficult to read in parts 

We tried our best to improve the quality of English language with the help of native speaker.

* More synthesis and analysis in both the introduction and results. At the moment, both sections largely consist of lists of relevant aspects of other papers, without an overall arching story. 

Thank you very much for your valuable comments . The manuscript is almost rewritten and result are synthesized again

* More detailed description of methods. For example, what specific search terms and combinations of these were used? Which authors screened and analysed which papers? How exactly were papers analysed? 

More details in the methodology is added.

* Conclusions and recommendations do not flow logically from the results. Please make the ideas in your conclusion more explicit in the results section of the paper. 

This portion is re-written.

Very grateful to you for your time for report. Your comments really improve this manuscript.

Best Regards

Reviewer 2 Report

Title is very vague and is not representative of the review. Abstract: Expanded forms of abbreviations missing. Add the expanded form before using the abbreviations Abstract is not complete representative of the systematic review. Lacks proper methodology. Which databases were searched? What was the inclusion and exclusion criteria for the included studies? Add relevant details in the abstract. Major revision for Abstract needed in line with the review conducted. Methods: Provide justification for selecting articles from 1990 to 2020. Why not before and after this time frame? Methodology seems confusing. It’s not clear what type of review the authors are aiming at. Scoping Review? Systematic Review? Meta-Review? The authors cannot include everything into a systematic review “Systematic review, original articles and meta-analysis of peer-reviewed journal” With this methodology, the authors cannot synthesize all available evidence and produce an unbiased and reliable summary of the existing Pharmacy Services for Mental Health. There is no information/details regarding as to how the authors did the following essential aspects of a systematic review “Data extraction, Quality assessment, Data synthesis and analysis, Assessment of reporting bias” The authors need to rework their methodology and perform the review again with specific review criteria. Conclusion The conclusions drawn in the paper are not consistent with the evidence and arguments presented. Overall, the authors need to rework their methodology and perform the review again. The authors should use a robust methodology to conduct the systematic review.

Author Response

Very grateful to you for your time for report. Your comments really improve this manuscript.

Title is very vague and is not representative of the review.

Title is changed

 Abstract: Expanded forms of abbreviations missing.

Add the expanded form before using the abbreviations

Expanded forms is added

Abstract is not complete representative of the systematic review.

Abstract is re-written.

 Lacks proper methodology. Which databases were searched?

Included in abstract and main manuscript.

 What was the inclusion and exclusion criteria for the included studies

Listed in Table 1.

 Major revision for Abstract needed in line with the review conducted.

Included in last line.

 Methods: Provide justification for selecting articles from 1990 to 2020. Why not before and after this time frame? Methodology seems confusing. It’s not clear what type of review the authors are aiming at. Scoping Review? Systematic Review? Meta-Review? The authors cannot include everything into a systematic review “Systematic review, original articles and meta-analysis of peer-reviewed journal” With this methodology, the authors cannot synthesize all available evidence and produce an unbiased and reliable summary of the existing Pharmacy Services for Mental Health. There is no information/details regarding as to how the authors did the following essential aspects of a systematic review “Data extraction, Quality assessment, Data synthesis and analysis, Assessment of reporting bias” The authors need to rework their methodology and perform the review again with specific review criteria.

All the details are included in section 3.Methodology is further improved.

Secondary studies are excluded and only primary literature of original research was included.

Time constrain is removed from method section and further work was done.

 Conclusion The conclusions drawn in the paper are not consistent with the evidence and arguments presented. Overall, the authors need to rework their methodology and perform the review again. The authors should use a robust methodology to conduct the systematic review.

Conclusion is rewritten.

Thanks and Regards

Reviewer 3 Report

The manuscript healthcare-2286826 is devoted to the actual social and scientific problem, namely шncreasing the role of pharmacist in modern medicine. The reviewed article is interesting for scholars and theme of the article meets the scope of the journal. Work is performed at sufficient scientific level and has good quality. The manuscript may be considered for publication after major revision in Healthcare. Prior publication of this manuscript following points needs to be addressed:

  • The abstract must reflect the basic results of the article. Should be rewritten.
  • It would be good to broaden the discussion in the context of a more detailed presentation of ways to practical implementation of obtained results.
  • The topic discussed by the authors is quite debatable and can have many directions for development. Therefore, the approach proposed by the authors may have some limitations. To avoid this problem, I propose to add separate section "Limitations and prospects for further improving pharmacy services for mental health".
  • Moderate English changes required. There are grammar/typing and orthographical errors in the manuscript.

My decision is major revision

Author Response

Very grateful to you for your time for report.  Limitations and prospects are separated in further subsections.  English language is edited by a native speaker.

Thanks and Regards

Round 2

Reviewer 2 Report

The authors have mentioned that "Critical appraisal of methodological quality and risk of bias assessment of included papers were undertaken independently by two reviewers" Kindly provide results of methodological quality and risk of bias assessment. The manuscript still needs revision.

Author Response

The authors have mentioned that "Critical appraisal of methodological quality and risk of bias assessment of included papers were undertaken independently by two reviewers" Kindly provide results of methodological quality and risk of bias assessment. The manuscript still needs revision.

Very grateful to you for your professional comment

The results are included in the result section

3.3. Risk of bias and quality assessment 

More over detailed in ( Supplimentary 1)

Reviewer 3 Report

Revised manuscript is suitable for publication.

Author Response

Revised manuscript is suitable for publication.

Thanks allot. Regards